# Comparison of the Accuracy of Ground Reaction Force Component Estimation between Supervised Machine Learning and Deep Learning Methods Using Pressure Insoles

**DOI:** 10.3390/s24165318

**Published:** 2024-08-16

**Authors:** Amal Kammoun, Philippe Ravier, Olivier Buttelli

**Affiliations:** 1PRISME Laboratory, University of Orleans, 12 Rue de Blois, 45100 Orleans, France; amal.kammoun@univ-orleans.fr (A.K.); olivier.buttelli@univ-orleans.fr (O.B.); 2Emka-Electronique Company, ZA du Patureau de la Grange, 41200 Pruniers-en-Sologne, France; 3Research Group Sport, Physical Activity, Rehabilitation and Movement for Performance and Health (SAPRèM), University of Orleans, 45100 Orleans, France

**Keywords:** insole pressure measurement, force plate measurement, GRF component estimation, supervised machine learning, deep learning, manual material handling, walking activities

## Abstract

The three Ground Reaction Force (GRF) components can be estimated using pressure insole sensors. In this paper, we compare the accuracy of estimating GRF components for both feet using six methods: three Deep Learning (DL) methods (Artificial Neural Network, Long Short-Term Memory, and Convolutional Neural Network) and three Supervised Machine Learning (SML) methods (Least Squares, Support Vector Regression, and Random Forest (RF)). Data were collected from nine subjects across six activities: normal and slow walking, static with and without carrying a load, and two Manual Material Handling activities. This study has two main contributions: first, the estimation of GRF components (Fx, Fy, and Fz) during the six activities, two of which have never been studied; second, the comparison of the accuracy of GRF component estimation between the six methods for each activity. RF provided the most accurate estimation for static situations, with mean RMSE values of RMSE_Fx = 1.65 N, RMSE_Fy = 1.35 N, and RMSE_Fz = 7.97 N for the mean absolute values measured by the force plate (reference) RMSE_Fx = 14.10 N, RMSE_Fy = 3.83 N, and RMSE_Fz = 397.45 N. In our study, we found that RF, an SML method, surpassed the experimented DL methods.

## 1. Introduction

The determination of the Ground Reaction Force (GRF) is crucial in biomechanical research. It allows for the calculation of human dynamics and kinematics [1]. Its applications are numerous such as in sports [2], ergonomics, or in medical applications. Most often, the determination of the GRF has been applied in gait conditioning. For example, it has been applied in injury prevention [3], rehabilitation [4,5,6], or motor dysfunction evaluation (idiopathic scoliosis in adolescents [7] and Parkinson’s disease [8,9]).

GRF determination can also be of interest in the field of ergonomics. It can be used to identify working conditions conducive to the onset of musculoskeletal disorders (MSDs). These periarticular disorders affect muscles, nerves, tendons, ligaments, and joints. Back injuries account for 39% of all such disorders [10]. These can be caused by activities such as maintaining posture and Manual Material Handling (MMH) tasks [11]. The inverse dynamic method is used to calculate the joint stresses resulting from MMH [12]. The estimate will be improved by incorporating GRF data. Also, GRF values play a crucial role in estimating the force exerted on the lumbar region [13,14,15,16,17], enabling the detection of low back pain during MMH tasks.

In addition, in static situations (the person is standing still), with or without a load, GRFs are important for determining if it is possible to ascertain the person’s weight. In a static case, the person’s weight equals the sum of the vertical forces (Fz) of both feet. When considering the weight of the load, the person’s weight equals the sum of the vertical forces (Fz) of both feet, minus the weight of the load.

We have described the importance of GRF for walking, MMH, and static situations. Currently, GRF components are assessed using force plates. These force plates are not mobile, making it challenging to evaluate GRF in real-life situations outside of a laboratory setting, such as during walking activities. Furthermore, the high cost and heavy weight of these plates make it difficult to determine GRF at any given location, as they need to be acquired in large numbers. Presently, a low-cost instrumented insole solution, equipped with pressure sensors, is capable of estimating only the vertical component (Fz) by employing a linear combination of the pressure sensors, each weighted according to its corresponding sensor surface area.

Deep Learning (DL) and Supervised Machine Learning (SML) methods aimed to determine the relationship between insole Plantar Pressure (PP) data and GRF components in 3D, including the medial–lateral component (Fx), the anterior–posterior component (Fy), and the vertical component (Fz). In [18], an Artificial Neural Network (ANN) was employed to estimate the Fy component. In [19], the GRF components were estimated using Linear Regression (LRG). Other studies [7,20,21,22] utilized ANN to estimate GRF components. In [20], the estimation of GRF components was evaluated also using Locally Linear Neuro-Fuzzy (LLNF) and LRG methods. The authors in reference [7] employed the three estimation methods introduced by Rouhani et al. [20] as well as the wavelet neural network method. In [23], the Fz and Fy components were estimated using Bidirectional Long Short-Term Memory (BLSTM) and LRG methods. In [24], GRF components were estimated using ANN, Least Squares (LS), and Support Vector Regression (SVR) methods. For the estimation of GRF components, there are works [7,18,19,20,21,24] that focused on walking activities, while Joo et al. [22] concentrated on golf activities. Jacobs et al. [21] expressed interest not only in walking activities but also in calf raising. Hoitz et al. [23] focused on running activities. The studies [7,18,20,21,22,23,24] suggest using DL methods such as ANN and BLSTM as the optimal approach for estimating GRF components compared to SML methods. However, this recommendation is made without testing various SML methods.

DL methods have certain disadvantages compared with SML methods. DL models are computationally intensive and time-consuming to learn [25]. As a result, DL model learning often requires expensive GPU servers [25]. In addition, to find the optimal DL model, several hyperparameters need to be calculated and tested, such as the optimizer, the number of hidden layers, and the batch size. To find the best model, these hyperparameters are tested in different combinations, each requiring the calculation and evaluation of a complete model. The higher the number of hyperparameters, the greater the number of combinations to be tested. In addition, SML models are easier to deploy than DL methods.

In other fields, in terms of performance, DL methods are not always better than SML methods. In [26], the authors compared the performance of Random Forest (RF), ANN, and SVR for tool wear prediction. The results showed that RF, which is an SML method, generated more accurate predictions than the ANN method. In another study [27], ANN, RF, and Gradient Boosted Machines methods were compared to predict carbon and nitrogen levels in soil in agriculture. The results indicated that the performance of the three methods varied depending on the dataset used. In most cases, the RF method yielded better outcomes than the ANN method. In [28], the authors compared the performance of ANN and SVR methods in the rainfall-runoff modeling of the Awash Belo Watershed in the Awash River Basin in Ethiopia. Both methods showed comparable performance.

In this study, our aim is to evaluate the accuracy of three DL methods (ANN, Convolutional Neural Network (CNN), and LSTM) and three SML methods (RF, SVR, and LS) in estimating GRF components for both feet during six activities: static activities both with and without carrying a 5 kg load, and normal and slow walking, as well as two MMH tasks: carrying a 5 kg load from bottom to top and vice versa and carrying a 5 kg load from left to right and vice versa. To the best of our knowledge, no study has evaluated the estimation of GRF components from PP data of an insole pressure for static activities and MMH tasks. This evaluation is of considerable importance in the field of ergonomics. This represents our first contribution. Furthermore, to our knowledge, the comparison of the estimation accuracy of these six methods has not yet been explored in the literature. We aim to investigate whether DL methods, as used by previous studies [7,18,20,21,22,23,24], yield better results compared to SML methods. This represents our second contribution.

To conduct the proposed study, it’s necessary to assess the estimation accuracy of each DL and SML method. This evaluation is conducted using standardized laboratory conditions using force plates. These force plates act as the reference for measuring the GRF components (ground truth).

Figure 1 illustrates the flow chart for estimating GRF components from PP, where PP presents the input data. The estimation accuracy is evaluated after DL/SML modeling. The process consists of two stages. The first stage involves training the model using the training dataset (depicted in red). In the second stage, the model’s performance is evaluated with the help of metrics using the test data set (depicted in green).

Each block of Figure 1 will be elaborated upon in the subsequent section.

## 2. Materials and Methods

### 2.1. Materials and Protocol

In our research, we employed two pressure insoles (Moticon, ReGo AG, Munich, Germany) fitted with 16 capacitive pressure sensors and two force plates (AMTI, Model: BMS600900, dimensions: 600 mm × 900 mm, Watertown, MA, USA). Figure 2 presents the location of the 16 pressure sensors along the insole. The GRF components from two force plates and the PP data from the Moticon insole were recorded at 100 Hz.

Our study involved nine healthy male subjects (weight: 77 ± 11.1 kg, height: 178 ± 4.2 cm, age: 47 ± 15 years). Those subjects had normal foot morphology and did not suffer from any specific foot pathologies such as varus or valgus. Before enrollment, the subjects were provided with comprehensive information regarding the study’s objectives and procedures, and they provided written consent, adhering to the ethical standards outlined in the Declaration of Helsinki [29].

They wore the Moticon insole inside their flat basketball-type shoes, which were of the same shoe size (42 EU). Preceding the experiment, each participant did three calibration exercises with the Moticon insole: a slow walk, standing still, and shifts of the body weight. The participants performed six distinct activities on the two force plates to obtain GRF data components for both feet (one force plate per foot, Figure 3, Figure 4, Figure 5 and Figure 6). These tasks encompassed walking steps, durations for both static situations, and trials for both MMH tasks (reported values represent the range of steps, durations, or trials among all subjects for both feet):(1)normal walk of 6 to 10 trials, resulting in (7–20 steps on the force plates) (Figure 3);(2)slow walk of 6 to 10 trials, resulting in (8–22 steps on the force plates) (Figure 3);(3)static situation (standing still) (2–5 s) (Figure 4);(4)static situation carrying a 5 kg load (static situation with CL) (2–5 s) (Figure 4);(5)carrying a 5 kg load from bottom to top and vice versa (bottom-top with CL) (7–18 trials) (Figure 5);(6)carrying a 5 kg load from left to right and vice versa (left-right with CL) (5–11 trials) (Figure 6).

For the walking situations, the subjects were not instructed to adjust their gait on the force plates (which were integrated into a treadplate) and they walked freely and continuously on them; going down and up the treadplate for each trial, following a trajectory as indicated by the black arrow in Figure 3. The force plate and the treadplate were at the same level without any offset and these alignments were regularly checked by a technician. The length of the force plate allowed two steps to be taken before reaching the force plate. Each trial included the steps performed on the force plates. Therefore, our dataset exclusively contained the steps performed on the force plates. A single step is defined from the instant of initial contact at the heel (On-Heel) to the moment of foot lift-off (Toe-Off).

For carrying load situations, we used a 5 kg load that was acceptable for both male and female subjects and lower than the national standard and that of the French labor code.

For normal walk, we asked subjects to walk as they naturally do without hurrying. For slow walk, we asked subjects to walk more slowly than for normal walking, after explaining what normal walking was. For static situation, we asked subjects to remain in a standing position, without moving any part of their body, with their arms at their sides. For static situation carrying a 5 kg load, we asked subjects to hold the 5 kg load in a standing position, with their elbows bent at right angles so as to position the load horizontally, keeping their arms at their sides. For carrying a 5 kg load from bottom to top and vice versa, we asked the subject, starting from the position defined in the static situation, to take the load located on the chair, lift it, and put it down on table in front of him, then repeat same sequence in the opposite order, without hurrying, and repeat the whole sequence and continue it for as long as he felt no difficulty and/or weariness. For carrying a 5 kg load from left to right and vice versa, we asked the subject, starting from the position defined in the static situation, to take the load from the end of the table on the right, lift it, move it to the other end of the table on the left and put it down, without moving the feet, then repeat the same sequence in the opposite direction, without hurrying, and repeat the whole sequence and continue it for as long as he felt no difficulty and/or weariness.

**Figure 3 sensors-24-05318-f003:**
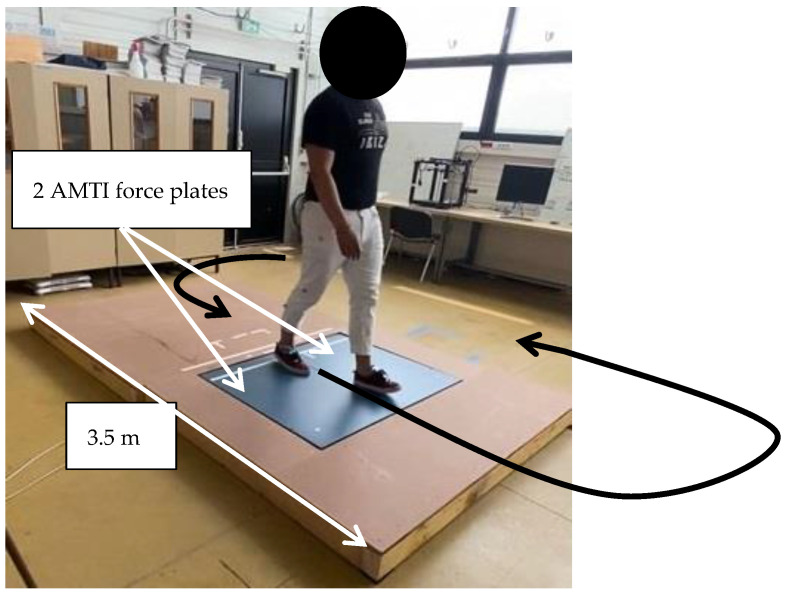
The subject goes down and up the plate for each trial following the black arrow trajectory for normal and slow walking.

In Figure 4, the subject stands still both without and with carrying a 5 kg load.

**Figure 4 sensors-24-05318-f004:**
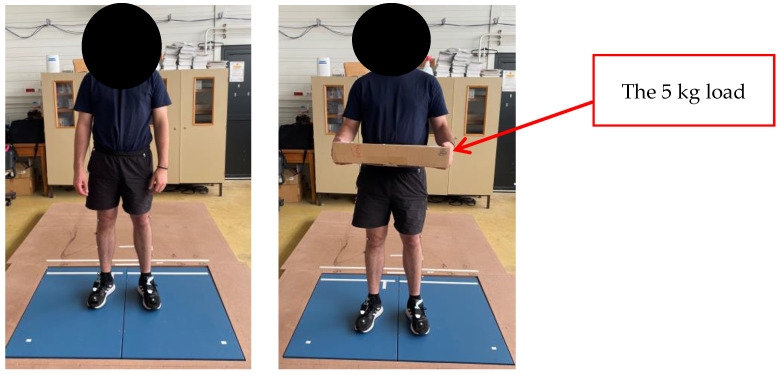
Static situation (**left** image), static situation with a 5 kg load (**right** image).

In Figure 5, the participant carries a 5 kg load from the bottom (chair: initial position) to the top (table) and vice versa.

**Figure 5 sensors-24-05318-f005:**
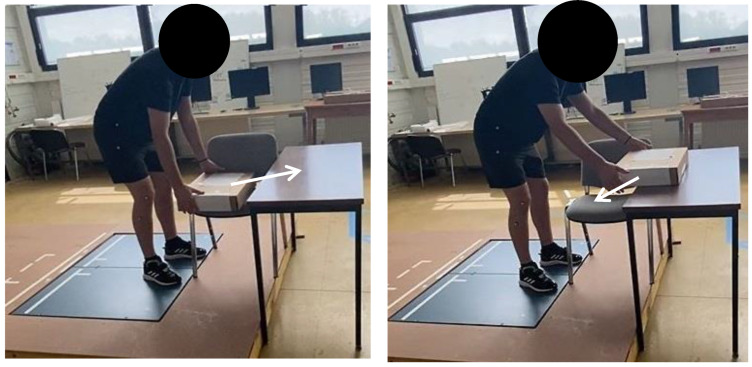
The participant starts by carrying a 5 kg load from the bottom (chair) (**left** image) to the top (table) (**right** image) and vice versa.

In Figure 6, the participant carries a 5 kg load from the far left of the table (initial position) to the far right of the table and vice versa.

**Figure 6 sensors-24-05318-f006:**
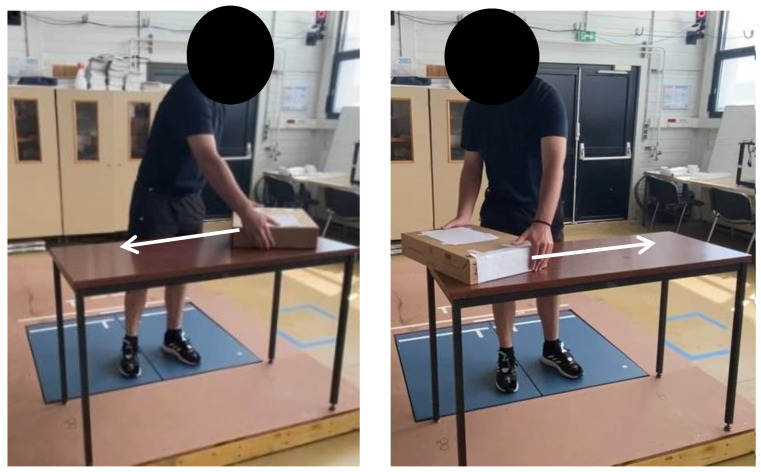
The participant starts by carrying a 5 kg load from the far left of the table (**left** image) to the far right of the table (**right** image) and vice versa.

### 2.2. Data Preprocessing

Linear interpolation of missing data of the insole

Some values were missing from the Moticon insole data. We filled these gaps using a linear interpolation.

Time-shift synchronization between the insole and force plate data

In the absence of a direct method for digitally or analogically synchronizing the data from the force plate with the insole data, we adopted a time-shift synchronization approach for both the right and left foot. We utilized the unique Fz component provided by the Moticon insole (where Fz is equivalent to the sum of PP multiplied by the sensor areas). Note that the length of the insole data is shorter than that of the force plate data. We took the Fz curve from the insole and shifted it relative to the Fz curve from the force plate over a given time range. Then, for each time shift, we calculated the Root Mean Square Error (RMSE) value and the correlation coefficient (R) as two functions of the time shift. The time shift value that yielded the lowest RMSE corresponds to the optimal moment for synchronizing the force plate data with the insole data. A high R value ensured a strong correlation between the two curves, thereby affirming the chosen time shift value. Figure 7 illustrates an example of synchronization for an excerpt of walking activity with the Moticon insole for the right foot.

Deletion of the data outside the force plate for the two walking activities

For the two walking activities, the subject performed trials on the force plate (Figure 3). We had samples of steps of the insole taken outside the force plate. We eliminated these samples and considered only those performed on the force plate (Figure 7).

### 2.3. Determination of the Optimal Architecture and Parameters of SML and DL Methods for GRF Component Estimation

To determine the optimal architecture and parameters of SML and DL methods for GRF component estimation, we conducted tests with different parameters and architectures for ANN, LSTM, CNN, RF, and SVR for the right foot. For more details on the DL methods for ANN, LSTM, and CNN, please refer to [23,24,30,31], respectively.

Artificial Neural Network (ANN)

(1)initialization: 1 input layer with 16 neurons, 2 hidden layers of (256, 128) neurons, activation function: sigmoid, normalization method for input (PP) and output (GRF components) data: mean, optimizer: Adamax, batch size: 32, learning rate: 0.01;(2)modify the optimizer: Adagrad, AdamW, Adadelta, Adam, Adamax, NAdam, RMSprop, Stochastic Gradient Descent (SGD) with different momentum values: 0, 0.5, and 0.9;(3)modify the learning rate: 0.04, 0.08, 0.01, 0.005, and 0.008;(4)modify the batch size: 1, 4, 16, 32, 64, 128, and 256;(5)modify the number of hidden layers (and their neurons): 1 layer: (150); 2 layers: (50, 50), (125, 125), (256, 128), and (128, 256); 3 layers: (256, 256, 128); 4 layers: (100, 100, 100, 100);(6)modify the activation function: tanh, leaky relu, softSign, relu, sigmoïde, wavelet, softPlus, and elu;(7)modify the normalization method: Min-Max in the range [0, 1] and [−1, 1], Mean, Z-Score, Robust Scaler, Vector Standardization, Maximum Linear Standardization, Decimal Scaling, Median, Tanh, Body Weight (BW), and Length Insole. These 12 normalization methods are explained in this study [24].

Long Short-Term Memory (LSTM)

(1)initialization: 1 input layer with 16 neurons, 1 LSTM layer with 128 cells, an ANN with 2 hidden layers of (128, 50) neurons, an input sequence size equal to 20 (the LSTM uses the PP data at the current time t as well as previous samples at times t − 19, …, t − 1 to estimate GRF components at the time t);(2)modify the number of LSTM layers (and their cells): 1 LSTM layer (128); 2 LSTM layers: (400, 200), (400, 100), and (800, 400);(3)modify the number of hidden layers of ANN (and their neurons): 1 layer (400); 2 layers (256, 128); 3 layers: (400, 100, 50) and (400, 300, 150);(4)test BLSTM instead of LSTM with the best architecture and parameters obtained from these 3 steps;(5)modify the number of BLSTM layers (and their cell): 1 layer (400); 2 layers: (400, 200) and (600, 200);(6)modify the size of the input sequence: 20, 40, 60, 80, and 100.

Convolutional Neural Network (CNN)

(1)initialization: 1 input layer with 16 neurons, 3 one-dimensional convolutional layers (Conv1D): Conv1D (number of filters = 14, kernel size = 4), Conv1D (number of filters = 10, kernel size = 4), Conv1D (number of filters = 8, kernel size = 4), 3 pooling layers: AvgPool1d (kernel size = 2), an ANN with 1 hidden layer of (32) neurons;(2)test MaxPool1d instead of AvgPool1d;(3)modify the kernel size of the convolutional layers (Conv1D): 2, 3, 4, and 5;(4)modify the number of convolutional layers: 1, 2, and 3;(5)modify the number of filters of Conv1D: 4, 8, 14, and 20;(6)modify the number of hidden layers of ANN (and their neurons): 1 layer (96); 2 layers (96, 50); 3 layers: (96, 60, 25) and (96, 100, 50).

For these three DL methods, to prevent overfitting, the learning rate was divided by a factor of 10 when the loss function on the validation set failed to decrease for 10 consecutive epochs, while the training loss decreased. If the validation loss kept decreasing for 45 consecutive epochs or reached 1000 epochs, the training stops. 

To implement ANN, LSTM, and CNN models, we used Pytorch (v3.10.7) library and NVIDIA RTX A4500 GPU.

Support Vector Regression (SVR)

The purpose of this method is to determine the hyperplane function (which may be nonlinear) that maximizes the number of measured data within the decision boundary [24]. We conducted tests with both linear and Radial Basis Function (RBF) kernels, varying the parameters. In particular, we conducted tests for both the linear and RBF kernels, varying ξ = 15 and 20, along with different values of C, including 0.01, 0.1, 1, 10, 50, 100, 300 600, and 900. Furthermore, we explored a supplementary parameter γ specific for RBF kernel, testing values of 0.01, 0.1, 1, 10, 50, 100, 300, 600, and 900. The parameters ξ, C, and γ are explained in [24].

Least Squares (LS)

The Least Squares (LS) method is a regression technique that allows for finding linear equations relating the GRF components to the PP data for both feet. This method revolves around minimizing the quadratic criterion between the measured and estimated output quantities from the selected linear mathematical model [24].

Random Forest (RF)

Random Forest (RF) is used to solve classification and regression problems. The RF method consists of multiple decision trees [32]. We tested the RF method composed of T trees, where T varies according to the following values: T = [10, 50, 100, 500, 1000, 5000, 10,000].

RF, LS, and SVR methods were implemented using Python (v3.10.7) and an Intel Xeon Gold 5218R @ 2.10GHz CPU.

The optimal parameters and architectures for the ANN, CNN, BLSTM, SVR, and RF obtained from the simulations, which exhibited the highest accuracy in estimating GRF components for the right foot, are presented in Table 1. 

The CNN and BLSTM methods used the optimal parameters (batch size, optimizer, learning rate etc.) of ANN.

For the sake of simplification, when modeling SML and DL methods for the left foot, we utilized the same optimal parameters and architectures that were used for estimating GRF components for the right foot.

### 2.4. SML and DL Modeling

To evaluate the performance of the six SML and DL models for estimating GRF components for each activity, metrics were calculated between the estimated (by insole PP data) and measured (by force plate) GRF components of the test datasets of both feet. This evaluation employed both intrasubject (intras) and intersubject (inters) strategies. The metrics include the correlation coefficient (R) and RMSE. 

To develop the SML and DL models for estimating GRF components, we used the datasets of 8 subjects for both feet. For the training set, 70% of the datasets were used. For the validation set, 10% of the datasets were used exclusively of the ANN, BLSTM, and CNN models. The models were evaluated using the intras strategy, where the model was tested on the test datasets of the 8 subjects, which accounted for 20% of the datasets. Furthermore, we evaluated the generalization capacity of our models using the inters strategy, where all the remaining dataset from the 9th subject was utilized for evaluation purposes. Table 2 presents the number of samples (steps or trials) of the whole dataset for the 9 subjects.

For the SML and DL methods, both intras and inters strategies involved rotating the training and test datasets to ensure robust results, employing a leave-one-subject-out cross-validation approach. This means that we constructed 9 models for each SML and DL method.

### 2.5. Metrics

We employed Root Mean Square Loss (RMSE) and correlation coefficient (R) [24] to assess the accuracy of the model for estimating GRF components:(1)$$\mathrm{RMSE}=\sqrt{\sum_{i=1}^{n}\frac{({\overset{̑}{y}}_{i}-y_{i}{)}^{2}}{n}},\;[N]$$and
(2)$$R=\frac{\sum_{i=1}^{n}\left({\overset{̑}{y}}_{i}-\mu (\overset{̑}{Y}))(y_{i}-\mu (Y)\right)}{\sqrt{\sum_{i=1}^{n}({\overset{̑}{y}}_{i}-\mu (\overset{̑}{Y}){)}^{2}\sum_{i=1}^{n}(y_{i}-\mu (Y){)}^{2}}}.$$
where n is the length of data of GRF components, $y_{i}$ is the measured GRF component (Fx, Fyor Fzby the force plate at time i, ${\hat{y}}_{i}$ is the estimated GRF component by the insole, $Y=y_{1},\ldots ,y_{n}$ is the measured GRF components, $\overset{̑}{Y}={\hat{y}}_{1},\ldots ,{\hat{y}}_{n}$ is the estimated GRF component, $\mu (\overset{̑}{Y})$ is the mean of estimated GRF components, and μ(Y) is the mean of measured GRF components.

## 3. Results

The performance of the nine models of ANN, BLSTM, CNN, SVR, LS, and RF is assessed using RMSE and R metrics. Figure 8 and Figure 9 display the mean of these metrics for the test dataset for each GRF component estimation and each activity, for both feet and both strategies, for each DL and SML method. The optimal estimation results are achieved using the method, whether DL or SML, that produces the most accurate results with the lowest mean RMSE value and ensures a higher mean R value.

Rather than present voluminous tables listing the metrics for all activities for each GRF component for both feet according to strategy, we opted for a graphical representation to facilitate reading and analysis. For more details, Table A1 and Table A2 in Appendix A present the mean and standard deviation (SD) of the RMSE and R metrics.

Figure 10 presents the curves of Fz estimated for both feet and measured using the RF method for samples from the test dataset for the intras strategy for the static situation with CL and the static situation. It also displays the curves of the summation of Fz estimated and measured for both feet and the curves of the subject’s weight for the static situation or the subject’s weight plus an additional 5 kg for the static situation with CL.

We conducted an analysis to identify the most effective estimation method (SML and DL) for each activity among the six methods for estimating each GRF component for both feet using the two strategies.

For the right foot, upon analyzing Figure 8, we found that for the intras strategy and for Fz estimation, ANN was the most effective method for the “static situation with CL”, “bottom-top with CL”, and “left-right with CL” activities. SVR proved the most effective method for “normal walk”, CNN for “slow walk”, and RF for “static situation”. Regarding Fy estimation, RF was the most effective method for “normal walk”, “slow walk”, “static situation”, and “bottom-top with CL”, while BLSTM proved to be optimal for “static situation with CL” and ANN for “left-right with CL”. For Fx estimation, ANN was the most effective method for “normal walk” and “left-right with CL”, BLSTM for “slow walk” and “static situation”, CNN for “bottom-top with CL”, and RF was the most effective method for “static situation with CL”. 

For the inters strategy, RF proved to be the most effective method for all the activities across the three GRF component estimations.

For the left foot, upon analyzing Figure 9, we found that for the intras strategy and for Fz estimation, ANN was the most effective method for “normal walk”, “static situation with CL”, “bottom-top with CL”, and “left-right with CL”. RF was shown to be the most effective method for “static situation” and SVR for “slow walk”. For Fy estimation, ANN was proven to be the most effective method for “normal walk” and “left-right with CL”, SVR for “slow walk”, BLSTM for “static situation”, LS for “bottom-top with CL”, and RF for “static situation with CL”. Regarding Fx estimation, ANN was the most effective method for “bottom-top with CL” and “left-right with CL”, BLSTM for “static situation with CL”, SVR for “normal walk”, and RF for “static situation” and “slow walk”. 

For the inters strategy, RF emerged as the most effective method for all the activities across the three GRF components. For Fx estimation, SVR also proved to be the second most effective method for the “left-right with CL” activity.

After analyzing Figure 8 and Figure 9, RF was shown to be the most accurate estimation for the static situation, with mean RMSE values ranging from 1.35 N to 12.59 N for the three GRF components for both feet and strategies (the minimum value of 1.35 is the mean RMSE for Fy in the inters strategy for the right foot, the maximum value of 12.59 is the mean RMSE for Fz in the inters strategy for the left foot). In contrast, LS gave the least accurate results for normal walking, with mean RMSE values ranging from 17.22 N to 86.04 N (the minimum value of 17.22 is the mean RMSE for Fx in the inters strategy for the left foot, the maximum value of 86.04 is the mean RMSE for Fz in the inters strategy for the left foot).

In Figure 10, we used the RF method to present Fz curves for both feet in the static situation with and without CL because it shows the best results. The objective of this Figure is to provide an example demonstrating that the subject’s weight in the “static situation” or the subject’s weight plus an additional 5 kg in the “static situation with CL” is equal to the summation of Fz for both feet. 

In the case of the “static situation with CL”, there is a small difference between the curves of the summation of Fz measured for both feet and the subject’s weight plus an additional 5 kg, varying from 5 to 14 N. This variation implies that external forces are added to the Fz force. These external forces are due to its movements, which can include posture adjustments and rocking movements to maintain balance (the subject carrying the load is not totally immobile).

## 4. Discussion

Based on the optimal mean R values from Figure 8 and Figure 9, our study indicated that, across both strategies, both feet, and six activities, the vertical component (Fz) of GRF (R_Fz = 0.878–0.994) could be estimated more accurately compared to the anterior–posterior component (Fy) (R_Fy = 0.459–0.876) and the medial–lateral component (Fx) (R_Fx = 0.538–0.950). This higher range of correlation values for estimating the Fz component could be attributed to DL- and SML-constructed models that better described the relation between the Fz component and the pressure sensors. This can be explained by the linear physical relationship between the Fz component and the pressure sensors (Fz is equal to the sum of the pressure sensors multiplied by the sensor area), which makes modeling much simpler than for other components.

In the literature, the authors in [18] used ANN to estimate Fy and found R_Fy = 0.621–0.963. The authors in [19] used LRG to estimate GRF components and found R_Fx = 0.719, R_Fy = 0.928, and R_Fz = 0.989, 0.992. The authors in [20] used ANN, LRG, and LLNF to estimate GRF components and found R_Fx = 0.764–0.937, R_Fy = 0.906–0.984, and R_Fz = 0.952–0.992. The authors in [7] used ANN, WNN, LRG, and LLNF to estimate GRF components and found R_Fx = 0.730–0.930, R_Fy = 0.878–0.979, and R_Fz = 0.921–0.993. The authors in [22] used ANN and WNN to estimate GRF components and found R_Fx = 0.730–0.900, R_Fy = 0.650–0.940, and R_Fz= 0.650–0.980. The authors in [23] used BLSTM and LRG to estimate Fy and Fz components and found R_Fy = 0.800–0.960 and R_Fz = 0.900–0.980. The authors in [24] used ANN, SVR, and LS to estimate GRF components and found R_Fx = 0.634–0.888, R_Fy = 0.606–0.675, and R_Fz = 0.952–0.979. This confirms the conclusion that the R values of the Fz component are higher than those of the Fy and Fx components, irrespective of the estimation method and the technology of the insole and the force plate.

We concluded that for estimating GRF components for both feet and both strategies, RF was the most effective method, followed by ANN, SVR, BLSTM, CNN, and finally, LS, in decreasing order of effectiveness (Figure 8 and Figure 9). These effectiveness rankings were based on 46, 14, 5, 5, 2, and 1 optimal configurations among the 72 total configurations. These configurations included three components, six activities, two feet, and two strategies. The RF method showed superior results compared to DL methods, particularly for the inters strategy for both feet. Contrary to the suggestions from previous research [7,18,19,20,21,22,23,24], our findings indicated that DL methods may not always be the optimal method for estimating GRF components compared to SML methods. Additionally, BLSTM (a very complex method to implement) and SVR (a simpler method) gave similar performances.

SML methods not only provide good results compared to DL methods but also offer other benefits. They frequently require less time for training to find the optimal model (see Section 2.3) and are more robust to variations in hyperparameters due to their fewer number. Adjusting hyperparameters in DL methods can be a challenge and a significant computational burden. In our study, training DL methods required the use of a GPU, while for SML methods, a CPU sufficed.

Figure 8 and Figure 9 suggested that the accuracy of each activity depends on the SML and DL methods (RF, SVR, LS, ANN, BLSTM, and CNN), as well as the strategy (intras or inters) used to estimate GRF components for both feet. Generally, for estimating GRF components across all six activities, both strategies and both feet, we recommended using the RF method, which exhibited mean RMSE values ranging from 1 to 1.4 times greater than the optimal results (the maximum value of 1.4 is the ratio between the mean RMSE of RF (26.78 N) and that of ANN (19.14 N) for “bottom-top with CL” for Fz in the intras strategy for the right foot). Appendix B presents the curves of the three GRF components for a “slow walk” step and a trial of “bottom-top with CL” and “left-right with CL” of the test dataset for the right foot using the RF method for the intras strategy.

In most cases, whatever the DL and SML method, whatever the foot, and whatever the strategy, the “static situation with CL” and “static situation” yielded the best results compared to “bottom-top with CL” and “left-right with CL”, which in turn yielded better results than “normal walk” and “slow walk”. In other words, the two static situations exhibited higher accuracy than the dynamic situations (MMH and walking), which is consistent because estimating GRF components in static cases is much easier (the variations of PP and measured GRF components by the force plate are small) than in dynamic cases. 

Note that the accuracy of GRF components may highly depend on the strategy employed. For example, ANN yielded good estimation results for intras strategy across all three components and the six activities for both feet, with mean RMSE values ranging from 1 to 1.26 times greater than the optimal results (the maximum value of 1.26 is the ratio of the mean RMSE of ANN (4 N) and that of BLSTM (3.18 N) for the “static situation with CL” for Fy in the intras strategy for the right foot). Conversely, the ANN method produced the worst results for the inters strategy, with mean RMSE values ranging from 1.42 to 12.02 times greater than the optimal results (the minimum value of 1.42 is the ratio of the mean RMSE of ANN (10.85 N) and that of RF (7.64 N) for “bottom-top with CL” for the right foot for Fx in the inters strategy for the left foot, the maximum value of 12.02 is the ratio of the mean RMSE of ANN (16.23 N) and that of RF (1.35 N) for “static situation” for Fy in the inters strategy for the right foot). This showed a lack of robustness and generalization capacities of ANN modeling. This problem could be partially solved by adding additional subjects and increasing the number of trials per subject [20].

## 5. Conclusions

We compared the accuracy of three DL and three SML methods for estimating GRF components for both feet for six activities. Many studies [7,18,20,21,22,23,24] recommend using DL methods for GRF component estimation without testing SML methods. Our study showed that DL methods are not always the best choice for GRF component estimation. The RF method yielded better results than the three DL methods. Additionally, the RF method require less hyperparameter tuning compared to DL models. It’s quicker and easier to find the best RF model, as there are not several combinations of hyperparameters to test. Also, implementing an RF model is simpler than the DL method.

However, our study is restricted to normal foot morphology, emphasizing the need for further research concerning other foot characteristics. In other words, for other conditions, it is essential to reassess the performance of the three DL and three SML methods in estimating GRF components.

In future work, our goal is to identify the important pressure sensors from the whole sensors of the insole, employing selection methods such as PCA. This selection process aims to decrease the number of sensors while preserving a high level of accuracy in estimating the GRF components.

## Figures and Tables

**Figure 1 sensors-24-05318-f001:**
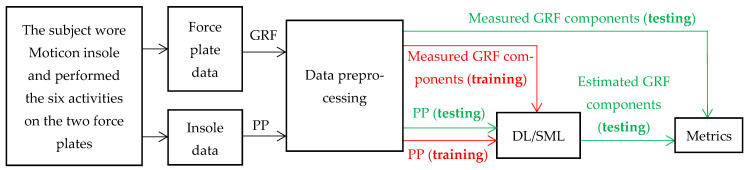
Flow chart for GRF component estimation from PP data. PP training dataset is utilized for DL/SML modeling, incorporating corresponding GRF force plate data (depicted in red). PP testing dataset is then employed to evaluate the performance of GRF components using the corresponding GRF force plate data (shown in green).

**Figure 2 sensors-24-05318-f002:**
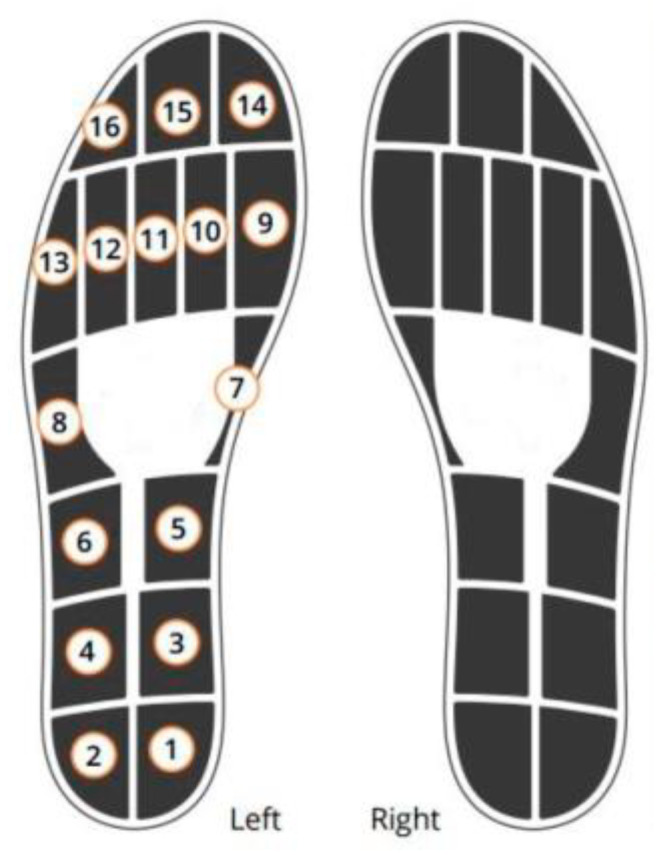
The location of the 16 pressure sensors along the insole.

**Figure 7 sensors-24-05318-f007:**
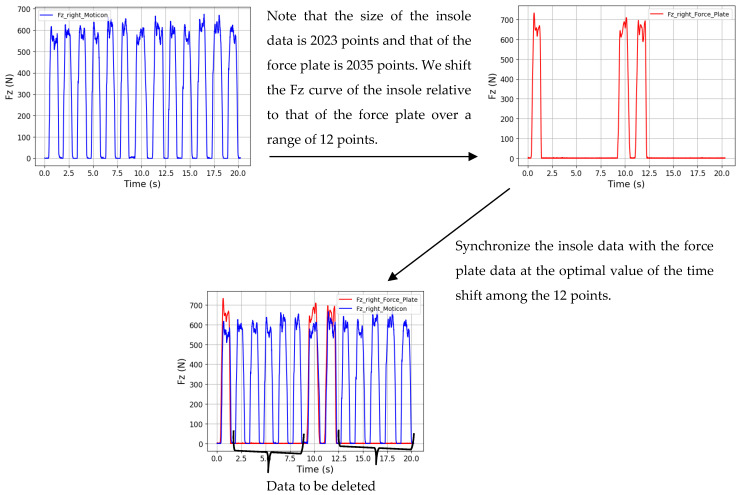
An example of synchronization for an excerpt of walking activity with the Moticon insole for the right foot.

**Figure 8 sensors-24-05318-f008:**
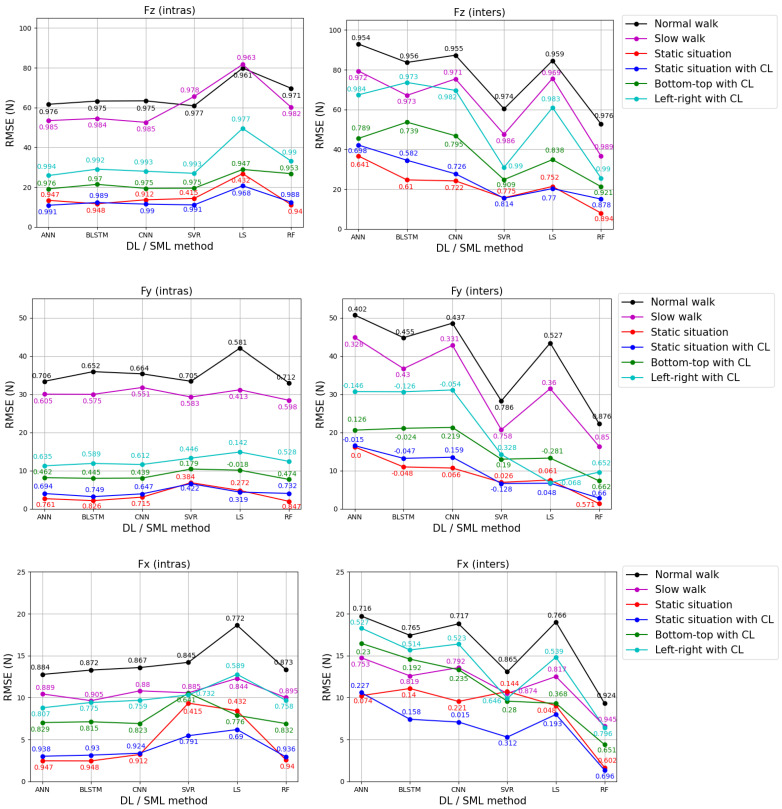
The curves of mean RMSE between the estimated (by insole PP data) and measured (by force plate) GRF components of the right foot for each DL and SML method for the test dataset, covering both strategies and each activity. The black curve represents the “normal walk”, the magenta curve represents the “slow walk”, the red curve represents the “static situation”, the blue curve represents the “static situation with CL”, the green curve represents “bottom-top with CL”, and the cyan curve represents “left-right with CL”. The mean R values are indicated on the RMSE values for each DL and SML method.

**Figure 9 sensors-24-05318-f009:**
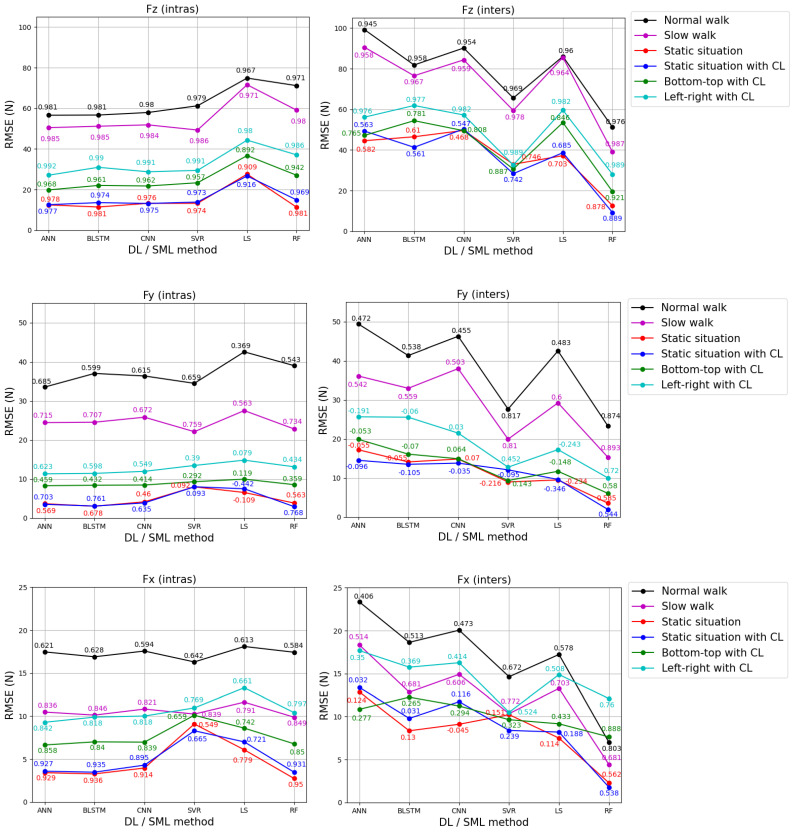
The curves of mean RMSE between the estimated (by insole PP data) and measured (by force plate) GRF components of the left foot for each DL and SML method for the test dataset, covering both strategies and each activity. The black curve represents the “normal walk”, the magenta curve represents the “slow walk”, the red curve represents the “static situation”, the blue curve represents the “static situation with CL”, the green curve represents “bottom-top with CL”, and the cyan curve represents “Left-right with CL”. The mean R values are indicated on the RMSE values for each DL and SML method.

**Figure 10 sensors-24-05318-f010:**
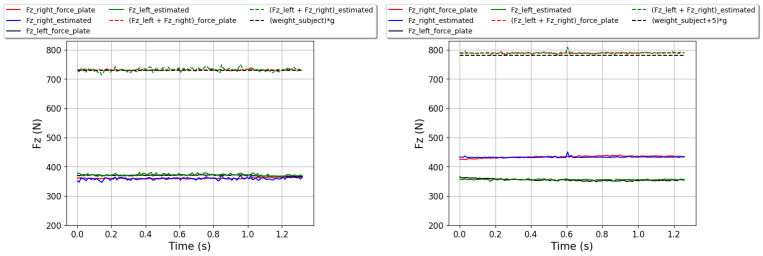
The curves of Fz estimated for both feet (the solid blue and green lines for the right and left foot, respectively) and measured (the solid red and black lines for the right and left foot, respectively) using the RF method for samples from the test dataset for the intras strategy. The curves of the summation of Fz for both feet estimated are presented by the dashed green line and those measured are presented by the dashed red line. The curves of the subject’s weight for the static situation or the subject’s weight plus an additional 5 kg for the static situation with CL are presented in the dashed black line. The right image pertains to the static situation with CL and the left image pertains to the static situation. In both images, g is the gravity, which is equal to 10 m/s².

**Table 1 sensors-24-05318-t001:** Optimal parameters and architectures of the SML and DL methods. Optimal parameters (batch size, optimizer, learning rate etc.) of ANN are applied for CNN and BLSTM.

ANN	CNN	BLSTM	SVR	RF
2 hidden layers of (256, 128) neurons, activation function: leaky relu, normalization method: BW, batch size: 4, optimizer: Adamax, learning rate: 0.01	1 convolution layer: Conv1D (number of filters = 8, kernel size = 4), 1 MaxPool1d layer (kernel size = 2), an ANN with 3 hidden layers of (96, 60, 25) neurons	2 BLSTM layers with (400, 200) cells, an ANN with 3 hidden layers of (400, 300, 150) neurons, an input sequence size equal to 20	RBF kernel model with ξ = 20, *C* = 100, and γ = 0.1	T = 1000

**Table 2 sensors-24-05318-t002:** The number of samples (steps or trials) of the whole dataset for the 9 subjects.

Activity	Right Foot	Left Foot
	Total Dataset: 160,183	Total Dataset: 164,526
Normal walk	11,100 (101)	12,719 (111)
Slow walk	15,008 (112)	17,732 (128)
Static situation	27,032	27,032
Static situation with CL	24,197	24,197
Bottom-top with CL	41,777 (86)	41,777 (86)
Left-right with CL	41,069 (72)	41,069 (72)

## Data Availability

Data unavailable for public sharing due to confidentiality.

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
