# Peer review of "Comparison of the Accuracy of Ground Reaction Force Component Estimation between Supervised Machine Learning and Deep Learning Methods Using Pressure Insoles"

_sensors, 2024, doi:10.3390/s24165318_

Round 1

Reviewer 1 Report

Comments and Suggestions for Authors

This paper shows the development of an algorithm that uses two insole sensors and two force plates to estimate the ground reaction force measured from the force plate, which is considered correct based only on the values ​​of the insole sensors. It includes a comparison of which performs better using several deep learning and supervised machine learning techniques.

The authors claim that the contribution of this study is the development of an algorithm to predict 3D GRF, and the development & comparative analysis of six algorithms. When looking only at the authors' introduction, it is considered to be original because it is a study that does not exist before. However, it appears that the reliability of the paper can be improved before it is published if the following corrections are reflected.

(1) If the appearance of the insole sensor used and location of the point for data acquisition (including 16 pressure sensors, information related to where each is located) are displayed in pictures or photos, it will help the reader understand the study and understand the following algorithm. It is believed that it will be more helpful in understanding.

(2) The experiment was conducted and data were acquired on 9 participants, but statistical verification of the reliability of the data composition is required. For example, when constructing a data set for a biomechanics experiment, it should be included that it has sufficient reliability because it produces a certain amount of statistical power when it is composed of several participants, several cases, and several trials. Among the presented contents, in the case of slow walking, it includes information on whether it is sufficient to proceed with 8-22 steps for 9 people (I wonder why it is different for each person) and whether the number of 15008 samples obtained here is appropriate for algorithm learning. I think this can improve the quality of the paper. If the dataset has insufficient statistical power, the contribution of this study is bound to be limited in “learning about various algorithms and recommending those that show good performance for each case.”

(3) Regarding the experimental environment, it seems that slow and normal walking were measured while turning counterclockwise in a narrow space. Since the front and back lengths are short, the motion or motor strategy to turn may be reflected in the ground reaction force. It seems like additional explanation is needed. And usually, the algorithm being developed needs to acquire data about the motion that will be mainly used, so it would be good to have additional explanation about why those motions were chosen. (For example, an explanation of what motion carrying a 5kg mass imitates)

(4) The second paragraph of the discussion contains a comparison of the results of this study with previous studies. However, it appears that the performance of the algorithm was compared using regression values ​​rather than comparing errors in force units. However, this may contain some errors. Static motion has relatively better performance at learning algorithms, but the more data of static motion is included, the higher the R is likely to be. Therefore, when comparing with other studies, it seems better to compare them by grouping them into the same category by motion and also comparing them in units of force.

Reviewer 2 Report

Comments and Suggestions for Authors

The work addresses the influence of pressure insoles on machine learning and deep learning. This topic can spark significant interest within the scientific community, but I believe the authors should consider the following points for its publication:

  • Remove the phrase " the authors in references " by “there are works that”.Line 71.
  • Footwear is an external element that can affect forces, especially absorption. It would be interesting to reflect whether all subjects wore the same footwear or to specify the type of footwear and its manufacture. Additionally, it is important for the authors to detail how the insoles were adapted to different foot sizes. What were the sizes of the subjects studied? Was there heterogeneity?
  • The population should be described.
  • It would be valuable for the authors to detail the placement or distribution of the sensors along the insole.
  • The authors should specify more concretely the instructions given to the subjects for performing the tests.

All the previous issues, as well as the fact that the platform was at different levels and unblinded, could bias the results. This should be reflected by the authors in the discussion.

Reviewer 3 Report

Comments and Suggestions for Authors

The authors have proposed the following manuscript: "Comparison of the accuracy of Ground Reaction Force component estimation between Supervised Machine Learning and Deep Learning methods using pressure insoles".

The authors compared the accuracy of GRF component estimation for both legs using six methods: three deep learning (DL) methods (artificial neural network, long-term memory, and convolutional neural network) and three supervised machine learning (SML) methods (least squares, support vector regression, and random forest (RF)). Data were collected from 9 subjects in six tasks: normal and slow walking, static walking with and without carrying a load, and two manual material handling tasks.

The Introduction part describes very well the importance of GRF for gait, MMH and static situations. GRF components are evaluated using force plates. The Deep Learning (DL) and Supervised Machine Learning (SML) methods were used to determine the relationship between plantar pressure (PP) data in the sole and the 3D GRF components, including the medial-lateral (Fx), anterior-posterior (Fy) and vertical (Fz) components.

To evaluate the performance of the six SML and DL models for component estimation for each activity, metrics were computed between the estimated (via the foot PP data) and measured (via the force plate) GRF components of the test datasets of both feet. Their evaluation used both intrasubject (intras) and intersubject (inters) strategies.

The authors concluded in their study that for both legs and six activities, the vertical component (Fz) of the GRF (R_Fz = 0.878 - 0.994) could be more accurately estimated compared to the anterior-posterior component (Fy) (R_Fy = 0.459 - 0.876) and the medial-lateral component (Fx) (R_Fx = 0.538 - 0.950).  The authors showed that both static situations showed a higher accuracy than the dynamic situations (MMH and walking), which is consistent because the estimation of GRF components in the static cases is much easier (the variations of PP and GRF components measured by the force plate are small) than in the dynamic cases.

The manuscript is very well written, structured and presented and I suggest that it be accepted for publication.

Round 2

Reviewer 2 Report

Comments and Suggestions for Authors

The authors addressed some of the considerations that were provided in the previous review, thus improving the work. 

However, some issues were not addressed in the depth that is considered necessary. Regarding footwear, the authors did not provide any additional information to the previous manuscript, implying that there is a heterogeneity that could bias the results in some way. The authors were asked to make a more in-depth description of the population studied, but they did not make the changes, as it appears in the manuscript, it is very superficial. Along the same lines, the authors did not address the previous comment: "It would be valuable for the authors to detail the placement or distribution of the sensors 

along the insole." . 

The challenges of comments can be considered to have been resolved.
